# Formalizing Attack Tree on Security Object for MySANi in Legal Metrology

**Muhammad Azwan Ibrahim** [1], **Faizan Qamar** [2,*], **Zarina Shukur** [2], **Nasharuddin Zainal** [3], **Nazri Marzuki** [1] and **Maria Ulfah Siregar** [4]

1. Electrical Group, National Metrology Institute of Malaysia, Sepang 43900, Malaysia
2. Center for Cyber Security, Faculty of Information Science and Technology, Universiti Kebangsaan Malaysia, Bangi 43600, Malaysia
3. Department of Electrical, Electronic and Systems Engineering, Faculty of Engineering and Built Environment, Universiti Kebangsaan Malaysia, Bangi 43600, Malaysia
4. Department of Informatics, Faculty of Science and Technology, UIN Sunan Kalijaga, Yogyakarta 55281, Indonesia
* Correspondence: faizanqamar@ukm.edu.my

**Abstract:** Illegal software manipulation is one of the biggest issues in software security. This includes the legally relevant software which are now crucial modules in weight and measuring instruments such as weighbridges. Despite the advancement and complexity of weight and measuring instruments, the inspection methodology is weak and lacks of innovation. The conventional inspection method is merely based on the observation printed certificate of the software. This paper introduces Malaysia Software-Assisted Non-Automatic Weighing Instrument (NAWI) Inspection (MySANI), a method used to enhance the software inspection scheme in legal metrology. MySANI introduces security objects in order to assist and enhance the inspection process. The security evaluation is based on the best practices in IT in metrology, where the attack model on relevant assets of the security objects is simulated for the Attack Probability Tree. The attack tree is verified by integrating formal notation and comparison with finite state transition system domain to verify the correctness properties of the tree design before the model can be further used in a risk analysis procedure within the Attack Probability Tree framework. Results show that the designed attack tree is consistent with the designed simulation.

**Keywords:** formal model; specification techniques; infrastructure protection; security; integrity; law

## 1. Introduction

Pattern approval (PA) is an examination and evaluation of regulated measuring instruments conducted by an impartial body to design an instrument prototype against national or international standards or statutory requirements [1]. This exercise is part of the legal metrology (LM) ecosystem to determine whether the measuring instrument is suitable for the intended use and can retain its accuracy and function under various environmental and operating conditions [2].

During the PA process, software evaluation, verification, and assessment are crucial for guaranteeing a credible and seamless operation of weighing and measuring equipment and systems [3]. The penalty for using forged measuring instruments is outlined in Section 17 of the Weights and Measures Act 1972 [4].

Despite the sophistication and complexity of measuring instruments, software enforcement framework within LM is poor. Furthermore, based on our literature studies, we found that there lack of best practices in tree structure and comprehensive tree design.

This paper introduces Malaysia Software-Assisted Non-Automatic Weighing Instrument (NAWI) Inspection (MySANI) which is a designed method in inspecting the compliance of universal computer-based software for NAWI within the LM framework in

Malaysia [5]. Security objects within MySANI are introduced, which is novel in the LM ecosystem. Thus, this proposed method enhances the overall inspection process within the LM framework in Malaysia. The paper demonstrates the use of SAND operator in the attack tree design and subsequently proving it using formal method against finite state transition which was impossible in the previous studies.

## 2. Related Works

In software engineering, risk analysis is one of the processes in risk assessment. It is a measuring stick for evaluating the effectiveness of security design of a system [6]. Around 50% of security problems are due to design weaknesses. Thus, the risk analysis performed at the design construction stage is very important for software security programs.

At the design stage, any risk analysis should be tailored to the software design [7]. Risks associated with threats can be modelled as risk = impact × likelihood.

Understanding the harm that an asset expose is necessary for effective asset protection. Risk assessment is used as a tool for information as an asset [8]. In information security, risk involves three main concepts: threat, vulnerability, and impact.

According to ISO/IEC 27005, the term risk is "combination of the consequences that would follow from the occurrence of an unwanted event and the likelihood of the occurrence of the event". Figure 1 shows the components in risk analysis.

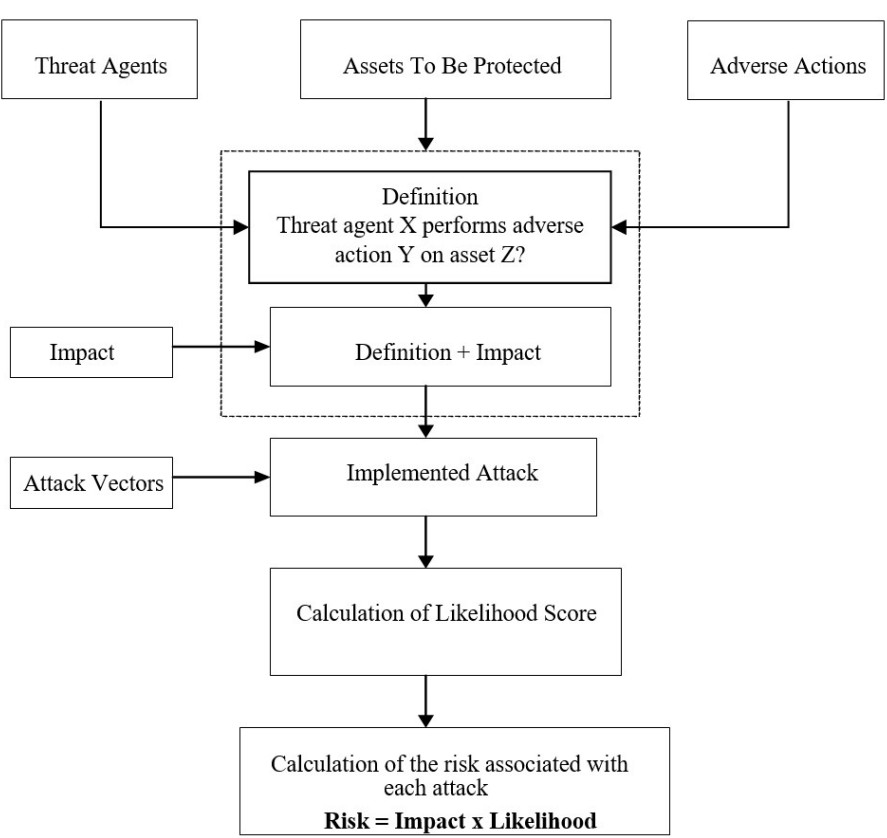

**Figure 1.** Component of risk analysis.

### 2.1. Security Evaluation in Legal Metrology

In LM, risk analysis is used as a tool to evaluate the security of a designed system or instrument. In a recent study, a method based on ISO/IEC 27005, 15408, and 18045 was developed by Esche and Thiel [9] to provide the foundation of security analysis in LM domain. However, this technique justifies the security of measuring instruments based merely on the device's technical features alone.

This method later was improved by incorporating the attacker's motivation. It was performed by integrating an attack tree modeling as part of the evaluation framework which is known as Attack Probability Tree (AtPT) [10].

*2.2. Attack Tree*

An attack tree is a model for visualizing and analyzing potential security threats [11]. This method has been adopted by the North Atlantic Treaty Organization (NATO) as well as the Open Web Application Security Project (OWASP) [12]. Several studies have been conducted by integrating attack trees with risk management methodologies such as supervisory control and data acquisition (SCADA), radio-frequency identification (RFID), and automated teller machine (ATM).

ATSyRA [13] made an approach by extracting a defined filter structure from an expert. This allows the semantic domain and structure of the filter to be found without intervention from experts. However, this method has imperfect (informal) label structure.

TREsPASS [14] applies the security knowledge base to the attack tree generated from the model system. However, the generic IT security knowledge base is incompatible for the LM situation.

Another formal design methodology is ADTool [15]. This method adopts the ADTree-based attack tree concept [16]. However, ADTree has a different approach with two types of players: strikers and defenders. This methodology, however, is not compatible with the AtPT scoring semantics used in the LM based on ISO/IEC 18045.

Attack tree design is a subjective procedure according to [17]. Esche in [10] emphasizes that in the field of LM, all measuring instruments share the same basic characteristics. Based on the fact that all instruments are based on the same technical solution it should be that the diversity of attack tree designs produce the same results.

Although the attack tree design is considered simple and intuitive, there is room for conceptual errors and a diversity of interpretations regarding the semantic refinement of the tree. Thus, there lack of best practices in tree structure and comprehensive tree design.

## 3. Research Methodology

Attack tree design in an AtPT framework is enhanced in this study where formal notation replaces the original version used only text to the name for each node [18]. We also improved the attack tree by adapting the SAND operator where the original model did not have it.

The attack tree decoration phases are formalized in mathematical notation in both the attack tree domain and finite state transition domain. This is to validate the correctness properties by using a method introduced by [19].

Table 1 shows the important notations that are used in the study.

**Table 1.** Notation with description.

| Notation | Description |
|----------|-------------|
| *Prop* | Set of propositions |
| $\iota$ | Initial configuration |
| $\gamma$ | Final configuration |
| $\mathcal{S}$ | State transition system |
| $S$ | Finite transition set |
| $s$ | Element/state |
| $\mathbb{N}$ | A set of natural numbers |
| $\lambda$: *Prop*$\to 2S$ | Labelling function |
| $\Pi, \rho$ | Path |
| $v$ | Attack vector |
| $T$ | Attack tree |
| $\alpha$ | Set of attributes |
| $\mathcal{R}$ | Risk scoring |
| $\langle \iota_n, \gamma_n \rangle$ | Leaf/goal |
| *OP* | Operator |

### 3.1. Finite State Transition

*Prop* is a set of propositions containing all the variables representing a situation at one time. This *Prop* is in the form of $\iota$, $\gamma$ to denote the state before ($\iota$) and after an event ($\gamma$). In this context the event is an attack vector $v$.

A state transition system is a tuple $\mathcal{S} = (S, \to, \lambda)$, where $S$ is the finite transition set (elements of $s$, $s_i$) for $i \in \mathbb{N}$ while $\to \subseteq S \times S$ is the transition relation to the system and assumed to be the left total. The $\lambda$: *Prop* $\to 2S$ is the labeling function. A state $s$ is labelled as $p$ when s $\in \lambda$ ($p$) and the size of $S$ is $|S| = |S| + |\to|$.

A path in system $S$ is the state condition where it has non-empty sequences of states. The symbols such as $\pi$ and $\rho$ are used to depict the paths. A cycle in paths $\pi \in \Pi(\mathcal{S})$ is a factor of $\pi'$ of $\pi$ where $\pi'(0) = \pi'(|\pi'|)$.

An elementary path is a path which has no cycle. A path $\pi$ which is elementary, does not have any state that more than once. Therefore, $|\pi| \leq |\mathcal{S}|$, removing all the cycles from $\pi$ iteratively until the resulting path is non-elementary. Table 2 summarizes the attack tree critical analysis.

**Table 2.** Attack tree critical analysis.

| Description | Esche | Audinot |
|-------------|-------|---------|
| Tree Model | generic | formal |
| Refinement | OR, AND | OR, AND, SAND |
| Node Name | informal, ordinary text-based | formal notation $\langle \iota, \gamma \rangle$ |
| Node Description | action-based | state-based |
| Design validation | unavailable | finite state transition |

### 3.2. Formalizing Attack Tree

In conjunction with the analysis of AtPT methodology [20], attack vector $v$ for attack tree $T$ over a set of attack vector attributes (Table 3) is the refinement for the path $\pi$ from with the direction from $\iota$ to $\gamma$. Assuming $v \in \to$, where $v_1, v_2, \ldots v_n$ is the attack vectors for each $T_1, T_2, \ldots T_n$, for $1 \leq i \leq n$. The size of attack vector is $|v| = |T| \ni \forall T_n \exists v_n$. The attack vector evaluation consists of conditions $OP \in \{OR, AND, SAND)$ and has the number of arity $n \geq 2$. The main goal of a an attack vector $v$ is $\gamma$ where the original state is $\iota$ and where the operator is $OP$. The attack tree can be represented in the form of $T = (\langle \iota, \gamma \rangle, OP)(T_1, T_2, \ldots T_n)$. Where the size of an attack tree $|T|$ is the number of attack vectors in $T$ such that $|\langle \iota, \gamma \rangle| = 1$ and $|(\langle \iota, \gamma \rangle, OP)(T_1, T_2, \ldots T_n)| = 1 + \sum_{i=1}^{n} |T_i|$.

**Table 3.** Attack vector attributes.

| Symbol | Attribute |
|--------|-----------|
| $Du$ | time required |
| $Ex$ | expertise |
| $Kn$ | knowledge required |
| $Wn$ | windows of opportunity |
| $Eq$ | equipment required |

Let the set of attributes $\alpha = \{Du, Ex, Kn, Wn, Eq\}$ for all $v_1, v_2, \ldots v_n$ be the attack vectors for $T_1, T_2, \ldots T_n$. While the $R$ be the AtPT risk scoring calculation based on the $OP$ over the set $\alpha$. Therefore, $v_i = Z\ (OP, \alpha)$ where, $OP \in \{OR, AND, SAND \mid 1 \leq i \leq n \}$.

*3.3. Attack Tree Correctness Property*

According to [21], an attack tree is considered to have the meet property when:

$$\llbracket OP(\langle \iota_1, \gamma_1 \rangle, \langle \iota_2, \gamma_2 \rangle, \ldots \langle \iota_n, \gamma_n \rangle) \rrbracket^S \cap \llbracket \langle \iota, \gamma \rangle \rrbracket^S \neq \varnothing$$

While undermatch property has:

$$\llbracket OP(\langle \iota_1, \gamma_1 \rangle, \langle \iota_2, \gamma_2 \rangle, \ldots \langle \iota_n, \gamma_n \rangle) \rrbracket^S \subseteq \llbracket \langle \iota, \gamma \rangle \rrbracket^S$$

And overmatch reflects:

$$\llbracket OP(\langle \iota_1, \gamma_1 \rangle, \langle \iota_2, \gamma_2 \rangle, \ldots \langle \iota_n, \gamma_n \rangle) \rrbracket^S \supseteq \llbracket \langle \iota, \gamma \rangle \rrbracket^S$$

Finally, the match property is:

$$\llbracket OP(\langle \iota_1, \gamma_1 \rangle, \langle \iota_2, \gamma_2 \rangle, \ldots \langle \iota_n, \gamma_n \rangle) \rrbracket^S = \llbracket \langle \iota, \gamma \rangle \rrbracket^S$$

In other words, the meet property is a minimum property for an attack tree before further analysis can be performed. This proves that at least one path can achieve the main goal and its refinements. This feature indicates that this tree model is essentially considered to be correct, and researchers can already begin to estimate the security of a system.

Undermatch shows that all paths that meet the refinement of a node also meet the goal itself. It can be considered as an underestimate for a set of scenarios and is enough to find the weaknesses of a system. While overmatch, on the other hand indicates that all paths that achieve the root node goal also met all decomposition to sub-goals. This is an assumption. If the tree characteristics are found to be a match, then it is considered that a tree has one hundred percent similarity with the system analyzed.

A tree should have a minimum of admissible and meet features for a tree design to be considered correct with respect to the analyzed systems.

**4. Malaysian NAWI Software Inspection for Computer**

Malaysia Software-Assisted Non-Automatic Weighing Instrument Inspection (MySANI) is a proposed method designed to enhance the inspection process within the framework of the LM ecosystem in Malaysia as depicted in Figure 2 [22]. It consists of two parts, which are PA and market surveillance.

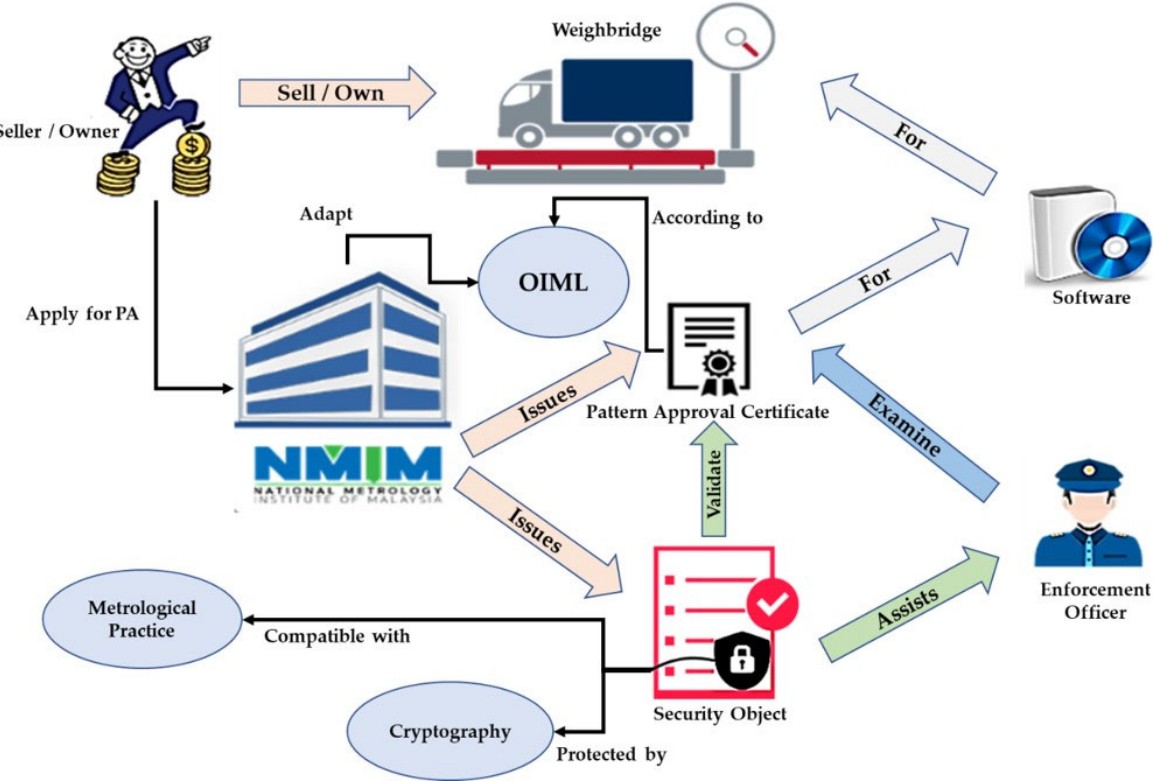

**Figure 2.** Contextual framework.

In Malaysia, PA is stated in the Weights and Measures Act 1972 (Act 71) and the National Measurement System Act 2007 (Act 675). The PA is provided for the National Measurement Standard Laboratory (NMSL) to carry out inspection and evaluation activities of measuring equipment [23,24]. Each measuring equipment has its own set of rules and laws enforced by specialized government enforcement agencies. For example, smoke meters, axel scales, and speed traps are enforced by the Road and Transport Department. Meanwhile measuring instruments for trade purposes are under the jurisdiction of Ministry of Trade and Consumer Affairs. PA covers all trade and law enforcement activities to ensure that the instruments meet the requirements of legal and international standards. It also ensures that measurement accuracy, fair trade, and law enforcement function well in all environmental conditions and situations [25]. The instrument was tested and evaluated from various physical aspects in a variety of conditions, including software functionalities [26].

Illegal usage of software for NAWI is detected in the market during market surveillance once the pattern is approved [27]. This activity is carried out by regular inspections such as by a government-appointed third-party organization (as in the current situation). There are three elements to be checked on NAWI software during market surveillance:

1.　Printed certificate/hard copy certificate (HC) information;
2.　Certificate–software pairing is correct;
3.　Legally relevant (LR) part of the software is intact.

If any of these stated elements are in doubt, the surveillance check is considered failed. The enforcement officer can take further steps such as sealing the whole system and seizing the whole measuring instrument system for further investigation.

The details of MySANI workflow during pattern approval and for market surveillance are depicted in Figures 3 and 4, respectively.

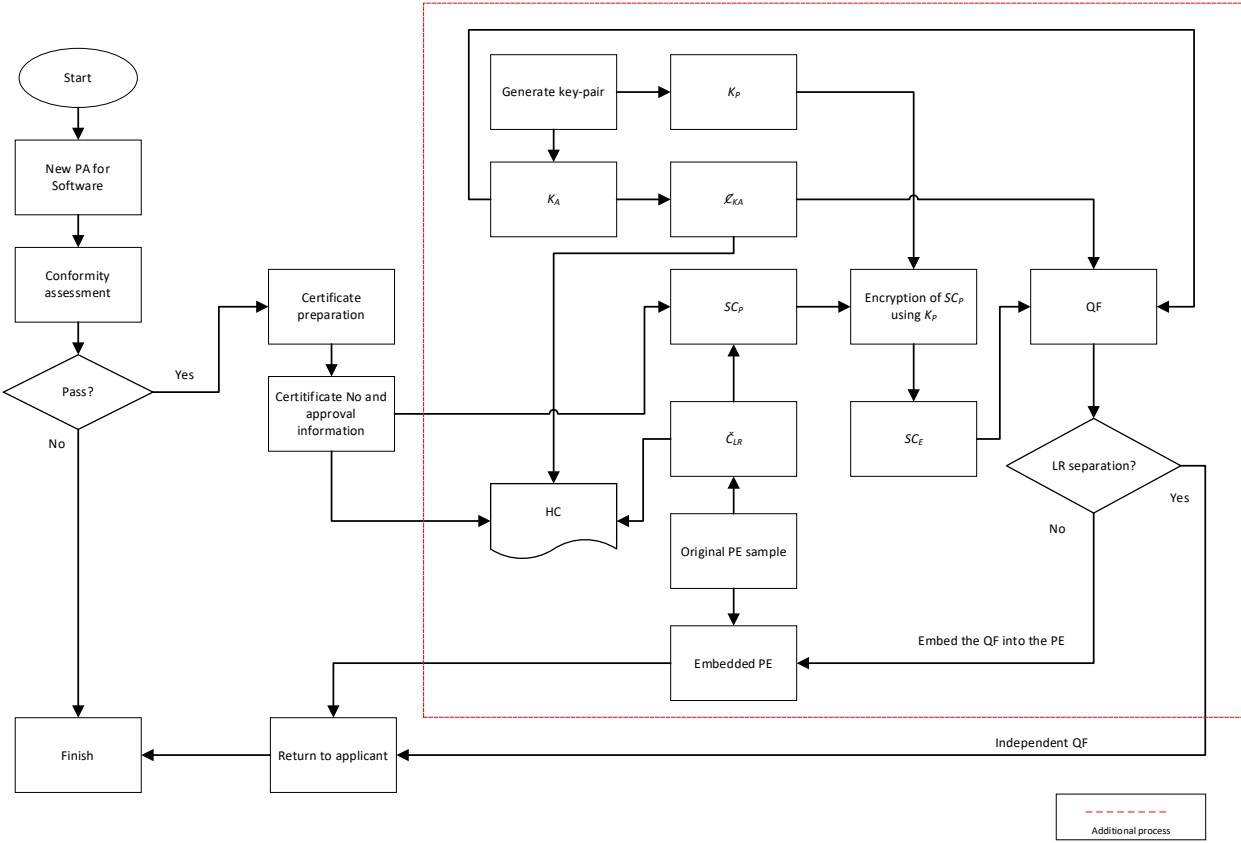

**Figure 3.** Proposed workflow during pattern approval.

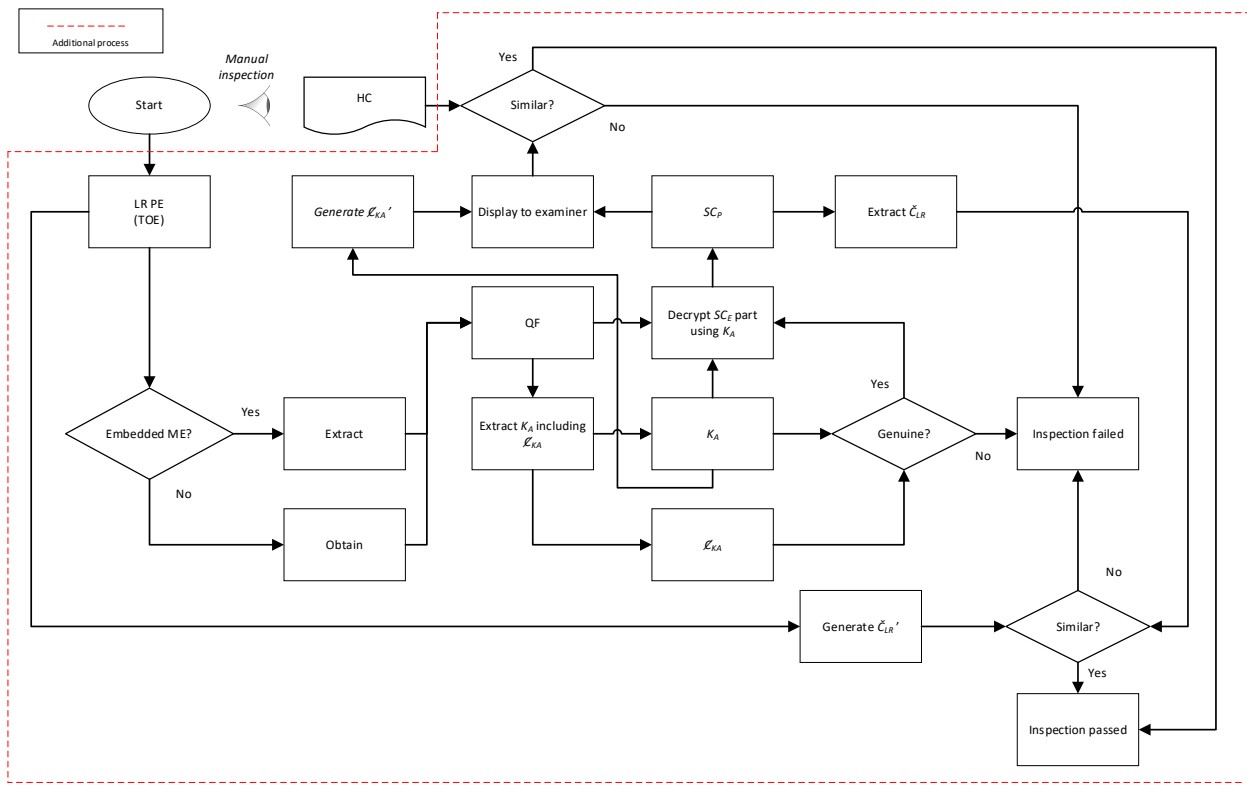

**Figure 4.** Proposed workflow during market surveillance.

### 4.1. Security Object

To achieve the aforementioned objectives, security objects (SOs) are implemented within the MySANI method. SOs consist of two parts: (1) soft-certificate plaintext ($SC_P$); and (2) quasi file (QF) [5]. Figure 5 illustrates the components of the security object which are used in MySANI.

1.  Transformed polynomial checksum for $\mathscr{C}_{KA}$;
2.  Plain checksum $\check{C}_{LR}$ for LR module which are recorded inside the $SC_P$.

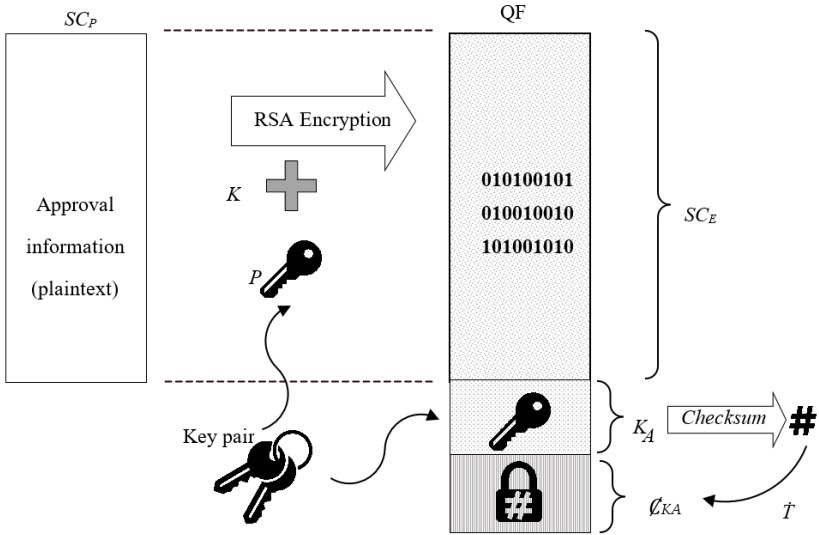

**Figure 5.** Security objects.

QF is designed as the security carrier to the $SC_P$. It protects the information contained from modification. Attacks on the $SC_P$ merely be focus on the QF security features. QF gathers several sections of data into single file. It consists of three main parts: ciphered soft certificate $SC_E$, public key $K_A$, and the checksum of $K_A$ with a hidden polynomial $\mathscr{C}_{KA}$. The $SC_P$ is encrypted using the RSA private key $K_P$ to become $SC_E$. $K_A$ and $K_P$ are generated by National Metrology Institute of Malaysia (NMIM) during certification process. $K_P$ is not disclosed and is only known by NMIM. The QF = {$SC_E$, $K_A$, $\mathscr{C}_{KA}$} consists of three sub-components as described in Table 4.

**Table 4.** QF components.

| Component | Description |
| --- | --- |
| $SC_E$ | Ciphertext of $SC_P$ |
| $K_A$ | Public Key (RSA) |
| $\mathscr{C}_{KA}$ | Checksum of $K_A$ which undergone polynomial transformation $\check{T}$ |

Let *decr* decryption and *encr* be the encryption function, $K_A$ as public key and $K_P$ as private key. $SC_E$ is the ciphered text of $SC_P$. Therefore, $SC_P = decr(SC_E, K_A) = decr(encr(SC_P, K_P), K_A)$ is the plaintext certificates which contain approval information that is important for inspection. It contains some of the plaintext information of static HC without security features $\forall SC_P : SC_P \subseteq$ HC where it uses an XML format based on the proposed DCC format [28].

### 4.2. Process Workflow

During the PA certificate preparation stage, the certificate number and approval of related information along with the important information are used in the preparation of the $SC_P$. The information in this is important and sufficient for field inspection purposes.

This $SC_P$ is then encrypted for the QF preparation process at the later stage. The checksum for the LR module $\check{C}$ is generated for the purpose of printed certificate HC preparation. Some important information is used in the preparation of the $SC_P$. The information in this is important and sufficient for field inspection purposes. This $SC_P$ is then encrypted for the QF preparation process at a later stage. Table 5 shows the information inside the plaintext certificate.

**Table 5.** Information inside plaintext certificate.

| Field | Description |
| --- | --- |
| Certificate owner | Name of approval |
| Approval number | Approval number |
| Approval mode | Mode of approval (full/conditional) |
| Date of approval | Effective date of approval |
| Validity period | The duration of approval validity |
| Software name | The official software name |
| LR module | List of LR module |

The MySANI workflow during pattern approval is shown in Figure 3. After a software passes the conformity assessment, while the certificate and approval information are being prepared for printed certificate HC, a pair of keys consisting of $K_A$ and private key $K_P$ are generated. The polynomial $\ell$ for $K_A$ is also calculated and placed inside HC. The soft certificate plaintext $SC_P$ contains approval information as well as the plain checksum for legally relevant module $\check{C}$. The $SC_P$ is then encrypted by using $K_P$ and becomes the ciphered certificate $SC_E$. The three elements ($SC_E$, $K_A$ and $\ell_{KA}$) are then combined to form the QF. The QF is passed to the applicant or undergoes an embedding process to its main executable (ME) depending on the approval mode for distribution.

A special tool, MySANI inspection software (MySANI-IS), is developed to assist the enforcement officer to verify the security objects. The MySANI first tries to locate the QF; whether it is embedded within the main executable ME or as a standalone QF file. If it is embedded, then extraction procedures are performed to pull out the QF out of the ME. The QF is then extracted into three parts: $SC_E$, $K_A$, and $\ell_{KA}$. The $\ell_{KA}$ is used to determine the key's integrity and authenticity before it is used to extract the $SC_E$ to form the $SC_P$. The approval information is then extracted from the $SC_P$ and is displayed to the examiner for inspection. The plain $\check{C}$ is used to determine whether the legally relevant part of the software is intact or not.

*4.3. Polynomial Transformation*

The polynomial transformation $\check{T} = \{St_1, St_2, \ell\}$ is an algorithm function to generate a checksum with hidden polynomial $\ell$ instead of a standard plain checksum $\check{C}$, for alignment with the LM requirements. $\check{T}$ is realized by first calculating the checksum of two concatenated strings ($St_1$ and $St_2$) to become the $\ell'$ and then performing bitwise XOR off the resulting checksum with secret bytes $\ell$.

$\check{C}_{KA}$ is the unknown plain checksum of $K_A$ where $\check{T}(K_A, S_S, \ell)$: $\check{C}_{KA} \rightarrow \ell_{KA}$. The $\ell_{KA}$ is in different form of $\check{C}_{KA}$ with equivalent strength of the checksum algorithm such that $\check{C}_{KA} \Leftrightarrow \ell_{KA}$. Any modification that changes $\check{C}_{KA}$ also changes the $\ell_{KA}$.

$\ell_{KA}'$ checksum is calculated over the concatenated string of $j = K_A \Vdash S_S$, where $S_S$ is secret string and $j$ contains both strings in the form of $st_1$ and $st_2$ where the $st_1$ is the string of $K_A$ while $st_2$ is the string $S_S$ such that $K_A \Vdash S_S = \{st_1 \, st_2: st_1 \in K_A, st_2 \in S_S\}$. Then, $\ell_{KA}' = crcf(j)$ and $\ell_{KA} = \check{C}_{KA}' \oplus \ell$. The $\ell_{KA}'$ can be retrieved back for integrity comparison purposes by $\ell_{KA}' = \check{C}_{KA} \oplus \ell$.

Information within the $SC_P$ is retrieved back and displayed to the inspector during market surveillance to confirm the correctness of the printed certificate information. Inspection is considered failed if the information displayed within $SC_P$ is not the same as the information on the printed certificate.

The checksum of the legally relevant software is generated ($\check{C}_{LR}{}'$) in real time for the purpose of comparison with the stored $\check{C}_{LR}$ to check the integrity of the legally relevant module. Let the *extr* be the QF extraction function such that $((\check{C}_{LR}{}' = \check{C}_{LR} \rightarrow extr(QF)) \wedge (\neg \check{C}_{LR}{}' = \check{C}_{LR})) \rightarrow$ FAIL!.

## 5. Modelling Attack Scenario for MySANI

The attack tree is used to visualize the attack vectors that an attacker might perform. In order to model an attack tree, an attack scenario is required to be established in the first place. The attack scenario describes how the attacker might access the protected information and try to fake the information within the SO.

In this simulation, the attacker's main target is to modify the content of QF. The attacker himself is the owner of the software and indirectly also owns the QF (worst case scenario). The attacker has unrestricted access to the target artifact. The attacker is assumed to already possess MySANI-IS used by the inspector as the second target of attack.

The attacker realizes that the $K_P$ is required to encrypt back the amended $SC_P{}'$. Since the key pair uses the RSA algorithm and the original $K_P$ is not placed in the inspection's software, the only way is by generating the fake private key pair of $K_P{}'$ and the fake public key of $K_A{}'$. $K_P{}'$ is used to re-encrypt $SC_P{}'$ while $K_A{}'$ is placed back into QF replacing the original $K_A$.

In order for the status of the QF to remain valid during the field inspection, the attacker must find a way to perform reverse engineering on the software used by the inspector to obtain the $\check{T}$ algorithm which protects the $K_A$ as part of QF. Suppose the attacker succeeds in breaking the $\check{T}$ algorithm: in that case, the attacker can recalculate the fake hidden polynomial $\ell'$ and once again, the attacker can reconstruct the fake QF (QF′) by reassembling the above three parts, namely $SC_P{}'$, $K_A{}'$ and $\ell'$. If this happens, the inspection is in valid status even if the QF's contents are changed. The inspector is not able detect the irregularities of the QF.

The analysis is started by studying the attack vectors on QF. An attacker will not carry out an attack motivated by damaging or destroying the availability of the QF or altering any QF structure as this causes the status check to fail and the enforcement officer may immediately seize the entire system for further investigation.

### 5.1. Attack Simulation

The root of the attack tree is $v_t$, when it is the ultimate goal for the attacker with intention to fake the QF with fabricated information and leave no trace of evidence during the market surveillance inspection. The tree consists of three main branches ($v_1$, $v_2$, and $v_3$). Attack vector $v_1$ is about obtaining the QF, attack vector $v_2$ is about modifying the content, while $v_3$ is about hiding the trace. Figure 6 shows the attack tree designed for the simulation. The details of each of attack vector is described in Table 6.

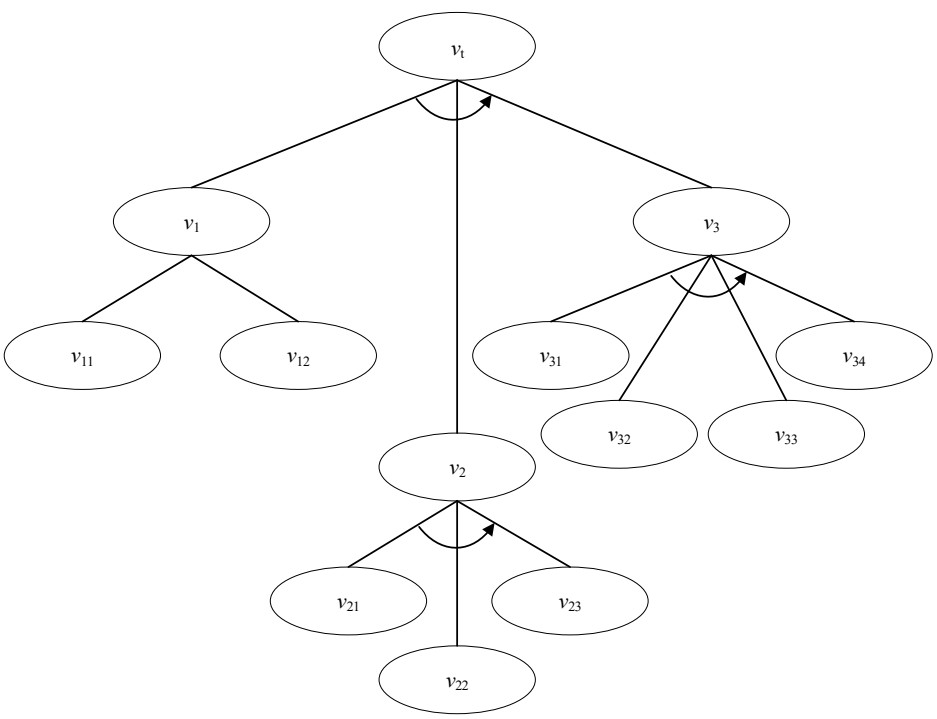

**Figure 6.** Attack tree analysis.

**Table 6.** Summary of attack vectors.

| Field | Description |
| --- | --- |
| $v_t$ | Attacker wants to fake the QF without any trace. |
| $v_1$ | Attacker wants to obtain the QF. |
| $v_{11}$ | The attacker extracts QF which is embedded inside the ME and tries to rebuild it into a single file. |
| $v_{12}$ | Attacker tries to obtain the QF which is already in a single file form. |
| $v_2$ | Attacker tries to amend the $SC_P$ information inside the QF. |
| $v_{21}$ | Attacker tries to obtain the $K_A$ which is part of elements in QF. |
| $v_{22}$ | Attacker decrypts the $S_{CP}$ using $K_A$. |
| $v_{23}$ | Attacker amends the information inside $SC_P$ into $SC_P'$. |
| $v_3$ | Attacker reorganize, reform the QF into fake QF'. |
| $v_{31}$ | Attacker generates fake $K_P'$. |
| $v_{32}$ | Attacker encrypts the $SC_P'$ using $K_P'$. |
| $v_{33}$ | Attacker tries to obtain the $\check{T}$ algorithm and recalculate the transformed polynomial based on $SC_P'$ and $K_A'$. |
| $v_{34}$ | The attacker reforms and rebuilds the QF' again using fake components. |

### 5.2. Security Object State Modelling

The formal validation method is performed by using a finite state transition system to validate the attack tree $T$ design model. Before analysis can be performed, the attack tree must also be presented in the form of formal notation.

$Q = \{Q^-, Q^+, Qg, Qt\}$: This variable specifies the state of QF. Where, $Q^-$ indicates the state where the attacker does not have QF and it is embedded in the ME target of evaluation (TOE), $Q^+$ indicates the state of the attacker does not have QF and is in the form of an individual file, $Qg$ is the state where the attacker already has QF and is ready for attack, and $Qt$ is the state where the attacker rearranges the QF with false elements.

$PKey = \{NaC, AcG, AcF\}$: This variable specifies the state of public key. $NaC$ indicates the attacker does not yet obtain the original public key $K_A$, $AcG$ indicates the original public key of the $K_A$ is successfully obtained, and $AcF$ the situation where the attacker generates and possesses a fake key pair.

$S_CP$ = {*Up*, *Oe*, *Oc*}: This variable indicates the condition of plaintext certificate. It consists of three states: Up is the situation where the SCP plaintext soft certificate is not opened, Oe is the situation where the SCP plaintext soft certificate is successfully opened, and *Oc* is the situation where the $SC_P$ plaintext soft certificate is amended with fake information.

*TpRev* = {*tt*, *ff*} is the variable which indicates the states of $\check{T}$ algorithm cracks, whether the attacker successfully cracks the algorithm (*tt*) or not (*ff*).

### 5.2.1. State Transition Modelling

Let the $Z$ system consist of two parties, namely the attacker and the QF. The system is modeled using a state variable, i.e., where its value determines the configuration possibilities for the system. In the initial settings, the attacker is assumed to not yet have a QF and intends to attack the QF by modifying the $SC_P$ without being able to be detected by the inspecting officer. To model the dynamic properties for the system, let the system be $(z_{i-1}, z_i) \in \rightarrow$, for every $1 \leq i \leq 9$. For the initial condition for the QF, $q = (Q = Q^-) \vee (Q = Q^+)$ while the $q' = (Q = Qg) \vee (Q = Qt)$ and $z_0, z_1 \in \lambda\,(q)$. For each state $z_i \in \lambda\,(q')$ for $3 \leq i \leq 9$.

The path $\rho$, shows the scenario for the attacker to execute an attack from the beginning of the situation to the end of the goal in the $\rho$ system. Table 7 summarized the state transition.

**Table 7.** State transition summary.

| Variable | $z_0$ | $z_1$ | $z_2$ | $z_3$ | $z_4$ | $z_5$ | $z_6$ | $z_7$ | $z_8$ | $z_9$ |
|---|---|---|---|---|---|---|---|---|---|---|
| $Q$ | $Q^-$ | $Q^+$ | **Qg** | Qg | Qg | Qg | Qg | Qg | Qg | **Qt** |
| PKey | NaC | NaC | NaC | **AcG** | AcG | AcG | **AcF** | AcF | AcF | AcF |
| ScP | Uo | Uo | Uo | Uo | **Op** | **Oe** | Oe | **Oc** | Oc | Oc |
| TpRev | ff | ff | ff | ff | Ff | ff | ff | ff | **tt** | tt |

For the transition system, only one variable changes at a time. $z_0$ indicates a situation where the QF is not yet owned, is embedded, and needs to be extracted into an individual file, the attacker does not have any cryptographic keys, while the content of the soft certificate are in its original form. The state can be represented as $Q = Q^-$ while the $PKey = NaC$, $z_0, z_1 \in \lambda\,(q)$ and $TpRev = ff$.

The transition is as follows:

- $z_1$ is the same as $z_0$ but QF is in the form of a separate file that needs to be identified by the attacker. The $Q = Q^+$ state transition occurs;
- $z_2$ is the same as $z_0$ but QF is successfully owned by the attacker. The $Q = Qg$ state transition occurs;
- $z_3$ is the same as $z_2$ but the attacker managed to obtain the $K_A$ public key from the QF element. The $PKey = AcG$ state transition occurs;
- $z_4$ is the same as $z_3$ but this time the attacker managed to open the $SC_P$ by decrypting the $SC_E$ using $K_A$. The $ScP = Op$ state transition occurs;
- the $z_5$ is the same as the $z4$ but the attacker generates a fake key pair that matches the original public key configuration. The following $PKey = AcF$ variable change occurs;
- $z_6$ is the same as $z_5$ but the attacker amended $SC_P$ to $SC_P'$. The $ScP = Oe$ variable values are changed;
- $z_7$ is the same as $z_6$ but the attacker encrypted $SC_P'$ using fake private key. The $ScP = Oc$ values are changed;
- $z_8$ is the same as $z_7$ except at this point the attacker successfully cracked the $\check{T}$ algorithm, $TpRev = tt$;
- And finally, $z_9$ is the same last state as the $z_8$ but the attacker reformed the QF with fake elements $Q = Qt$.

### 5.2.2. Attack Tree Formal Modelling

A similar attack is being modeled as attack tree $T$ for the system $\mathcal{Z}$ as discussed, which has a goal in the form $T = (\langle \iota, \gamma \rangle, \text{SAND})(T_1, T_2, T_3)$. Where the formal representation of each sub-tree:

$$T_1 = (\langle \iota_1, \gamma_1 \rangle), \text{OR})(\langle \iota_{11}, \gamma_{11} \rangle, \langle \iota_{12}, \gamma_{12} \rangle);$$
$$T_2 = (\langle \iota_2, \gamma_2 \rangle), \text{SAND})(\langle \iota_{21}, \gamma_{21} \rangle, \langle \iota_{22}, \gamma_{22} \rangle, \langle \iota_{23}, \gamma_{23} \rangle);$$
$$T_3 = (\langle \iota_3, \gamma_3 \rangle), \text{SAND})(\langle \iota_{31}, \gamma_{31} \rangle, \langle \iota_{32}, \gamma_{32} \rangle, \langle \iota_{33}, \gamma_{33} \rangle, \langle \iota_{34}, \gamma_{34} \rangle).$$

Each sub-tree $T_i$ has the atomic goal in the form of $\langle \iota_i, \gamma_i \rangle$. The same variables are applied to the state transition notation. Figure 7 shows the attack tree with formal notations.

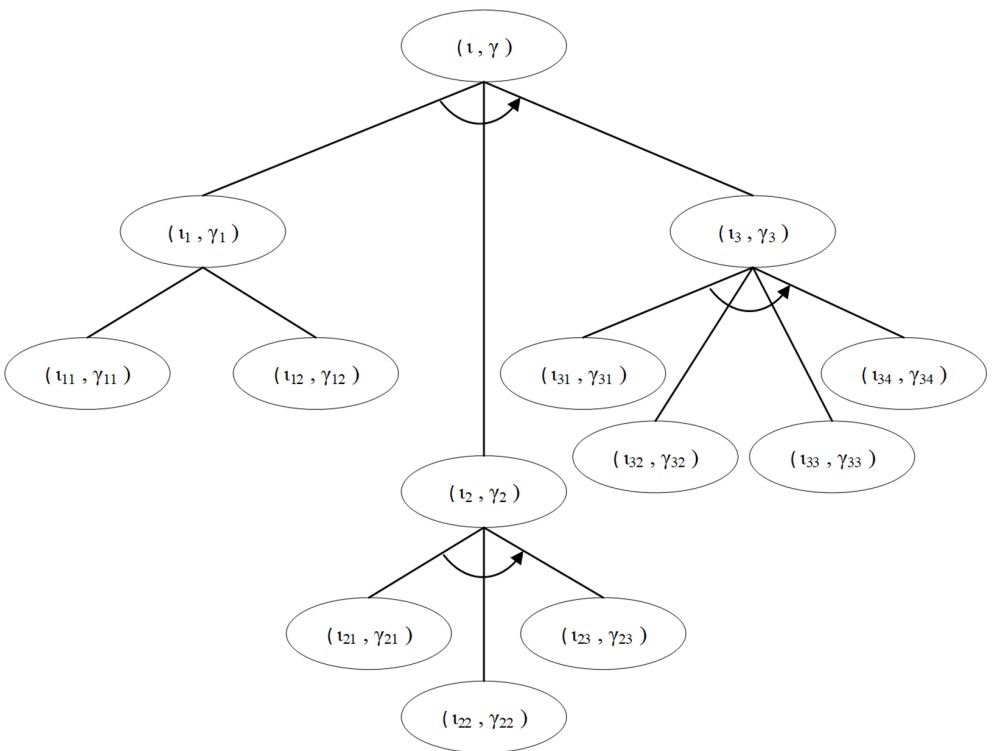

**Figure 7.** Attack tree with formal notations.

For the formal model the attack tree is as follows:

- The original situation is where the attacker does not yet have a QF, does not have a public key, is not able to access plaintext soft certificates, and did not successfully crack the $\check{T}$ algorithm.

$$\iota := (Q = Q^-) \wedge (PKey = NaC) \wedge (ScP = Uo) \wedge (TpRev = f\!f);$$

- The ultimate goal for the attacker is to be able to amend the plaintext soft certificate, possess a fake key pair, re-encrypt using a fake private key, and put a fake public key back in the amended QF.

$$\gamma := (Q = Qt) \wedge (PKey = AcF) \wedge (ScP = Oe);$$

- Atomic target $\langle \iota_1, \gamma_1 \rangle$: In the original state, the attacker does not yet have QF and needs to identify a QF as the target first. The goal is to have the QF in the form of individual file. Assuming $\top$ represents an empty configuration, therefore:

$$\iota_1 := \top \text{ and } \gamma_1 := (Q = Qg);$$

- Atomic goals $\langle \iota_{11}, \gamma_{11} \rangle$: QF is embedded in the ME and the attacker must first extract it into a single file. Therefore:

$$\iota_{11} := (Q = Q^-) \text{ and } \gamma_{11} = \gamma_1;$$

- Atomic goals $\langle \iota_{12}, \gamma_{12} \rangle$: QF is available as an individual file. The attacker needs to identify the location of the QF file. Therefore:

$$\iota_{12} := (Q = Q^+) \text{ and } \gamma_{12} := (Q = Qg);$$

- Atomic goal $\langle \iota_2, \gamma_2 \rangle$: The attacker aims to modify the $SC_P$ in the QF. Therefore:

$$\iota_2 := (Q = Qg) \wedge (PKey = AcG) \wedge (ScP = Uo) \wedge (TpRev = f\!f) \text{ and } \gamma_2 := \gamma;$$

- Atomic goal $\langle \iota_{21}, \gamma_{21} \rangle$: Attacker obtains $K_A$. Therefore:

$$\iota_{21} := (Q = Qg) \wedge (PKey = NaC) \text{ and } \gamma_{21} := (PKey = AcG);$$

- Atomic goal $\langle \iota_{22}, \gamma_{22} \rangle$: Attacker decrypts $SC_E$ using $K_A$. Therefore:

$$\iota_{22} := \gamma_{21} \text{ and } \gamma_{22} := (ScP = Op);$$

- Atomic goal $\langle \iota_{23}, \gamma_{23} \rangle$: Attacker modifies the contents of the $SC_P$. Therefore:

$$\iota_{23} := \gamma_{22} \text{ and } \gamma_{23} := (ScP = Oe);$$

- Atomic goal $\langle \iota_3, \gamma_3 \rangle$: The attacker aims to reconstruct the QF using modified plaintext data and fake keys. Therefore:

$$\iota_3 := (PKey = AcF) \wedge (ScP = Oe) \wedge (TpRev = f\!f) \text{ and } \gamma_3 := \gamma;$$

- Atomic goal $\langle \iota_{31}, \gamma_{31} \rangle$: Attacker generates fake $K_P{}'$. Therefore:

$$\iota_{31} := \top \text{ and } \gamma_{31} := (PKey = AcF);$$

- Atomic goals $\langle \iota_{32}, \gamma_{32} \rangle$: The attacker encrypts the $SC_P{}'$ by using fake key $K_P{}'$. Therefore:

$$\iota_{32} := (PKey = AcF) \wedge (ScP = Oe) \text{ and } \gamma_{32} := (ScP = Oc);$$

- Atomic goals $\langle \iota_{33}, \gamma_{33} \rangle$: Attack polynomial transformations based on $K_P{}'$ and $SC_P$. Therefore:

$$\iota_{33} := (TpRev = f\!f) \text{ and } \gamma_{33} := (TpRev = tt);$$

- Finally, the atomic goal $\langle \iota_{34}, \gamma_{34} \rangle$: The attacker reconstructs all elements into a false QF. Therefore: $\iota_{34} := (PKey = AcF) \wedge (ScP = Oe) \wedge (TpRev = tt)$ and $\gamma_{34} := \gamma_3$.

### 5.3. Formal Analysis and Discussion

$T_1$ is defined as $[\![\langle \iota_1, \gamma_1 \rangle]\!]^{\mathcal{S}}$ while for the sub-trees, are $[\![\langle \iota_{11}, \gamma_{11} \rangle]\!]^{\mathcal{S}}$ and $[\![\langle \iota_{12}, \gamma_{12} \rangle]\!]^{\mathcal{S}}$, respectively. Therefore, because of $[\![\mathrm{OR}(\langle \iota_{11}, \gamma_{11} \rangle, \langle \iota_{12}, \gamma_{12} \rangle)]\!]^{\mathcal{S}} = [\![\langle \iota_{11}, \gamma_{11} \rangle]\!]^{\mathcal{S}} \cup [\![\langle \iota_{12}, \gamma_{12} \rangle]\!]^{\mathcal{S}}$ while $[\![\langle \iota_1, \gamma_1 \rangle]\!]^{\mathcal{S}} \cap [\![\mathrm{OR}(\langle \iota_{11}, \gamma_{11} \rangle, \langle \iota_{12}, \gamma_{12} \rangle)]\!]^{\mathcal{S}} \neq \varnothing$, then the tree is considered to have the meet property.

Since the $[\![\mathrm{OR}(\langle \iota_{11}, \gamma_{11} \rangle, \langle \iota_{12}, \gamma_{12} \rangle)]\!]^{\mathcal{S}} \supseteq [\![\langle \iota_1, \gamma_1 \rangle]\!]^{\mathcal{S}}$, thus the $T_1$ is also has the overmatch property.

For $T_2$, it is defined as $[\![\langle \iota_2, \gamma_2 \rangle]\!]^{\mathcal{S}}$ and all the sub-trees can be refined as $[\![\mathrm{SAND}(\langle \iota_{21}, \gamma_{21} \rangle, \langle \iota_{22}, \gamma_{22} \rangle, \langle \iota_{23}, \gamma_{23} \rangle)]\!]^{\mathcal{S}}$; therefore, the sub-trees are equivalent to $[\![\langle \iota_{21}, \gamma_{21} \rangle]\!]^{\mathcal{S}} \cdot [\![\langle \iota_{22}, \gamma_{22} \rangle]\!]^{\mathcal{S}} \cdot [\![\langle \iota_{23}, \gamma_{23} \rangle]\!]^{\mathcal{S}}$. From a quick analysis, the $[\![\mathrm{SAND}(\langle \iota_{21}, \gamma_{21} \rangle, \langle \iota_{22}, \gamma_{22} \rangle, \langle \iota_{23}, \gamma_{23} \rangle)]\!]^{\mathcal{S}} \neq \varnothing$ and thus, $T_2$ is considered to have the meet property. More-

over, $[\![\mathrm{SAND}(\langle \iota_{21}, \gamma_{21}\rangle, \langle \iota_{22}, \gamma_{22}\rangle, \langle \iota_{23}, \gamma_{23}\rangle)]\!]^{\mathcal{S}} = [\![\langle \iota_2, \gamma_2\rangle]\!]^{\mathcal{S}}$; therefore, $T_2$ also has the match property.

For the last branch $T_3$, $[\![\langle \iota_3, \gamma_3\rangle]\!]^{\mathcal{S}}$ where the sub-trees can be refined as $[\![\mathrm{SAND}(\langle \iota_{31}, \gamma_{31}\rangle, \langle \iota_{32}, \gamma_{32}\rangle, \langle \iota_{33}, \gamma_{33}\rangle, \langle \iota_{34}, \gamma_{34}\rangle)]\!]^{\mathcal{S}}$ where it is equivalent to $[\![\langle \iota_{31}, \gamma_{31}\rangle]\!]^{\mathcal{S}} \cdot [\![\langle \iota_{32}, \gamma_{32}\rangle]\!]^{\mathcal{S}} \cdot [\![\langle \iota_{33}, \gamma_{33}\rangle]\!]^{\mathcal{S}} \cdot [\![\langle \iota_{34}, \gamma_{34}\rangle]\!]^{\mathcal{S}}$. By removing non-elementary paths $[\![\langle \iota_3, \gamma_3\rangle]\!]^{\mathcal{S}} \cap [\![\mathrm{SAND}(\langle \iota_{31}, \gamma_{31}\rangle, \langle \iota_{32}, \gamma_{32}\rangle, \langle \iota_{33}, \gamma_{33}\rangle, \langle \iota_{34}, \gamma_{34}\rangle)]\!]^{\mathcal{S}} \neq \varnothing$. Thus, the $T_3$ has the meet property and since $[\![\langle \iota_3, \gamma_3\rangle]\!]^{\mathcal{S}} = [\![\mathrm{SAND}(\langle \iota_{31}, \gamma_{31}\rangle, \langle \iota_{32}, \gamma_{32}\rangle, \langle \iota_{33}, \gamma_{33}\rangle, \langle \iota_{34}, \gamma_{34}\rangle)]\!]^{\mathcal{S}}$; therefore, $T_3$ also has the match property.

For the final path semantics of goal expression $T$, $[\![\langle \iota, \gamma\rangle]\!]^{\mathcal{S}}$, can be refined as $[\![\mathrm{SAND}(\langle \iota_1, \gamma_1\rangle, \langle \iota_2, \gamma_2\rangle, \langle \iota_3, \gamma_3\rangle)]\!]^{\mathcal{S}}$ and is equivalent to $[\![\langle \iota_1, \gamma_1\rangle]\!]^{\mathcal{S}} \cdot [\![\langle \iota_2, \gamma_2\rangle]\!]^{\mathcal{S}} \cdot [\![\langle \iota_3, \gamma_3\rangle]\!]^{\mathcal{S}}$. By removing non-elementary paths, similar to the previous sub-trees, $[\![\langle \iota, \gamma\rangle]\!]^{\mathcal{S}} \cap [\![\mathrm{SAND}(\langle \iota_1, \gamma_1\rangle, \langle \iota_2, \gamma_2\rangle, \langle \iota_3, \gamma_3\rangle)]\!]^{\mathcal{S}} \neq \varnothing$, thus the $T$ has the meet property and $[\![\langle \iota, \gamma\rangle]\!]^{\mathcal{S}} = [\![\mathrm{SAND}(\langle \iota_1, \gamma_1\rangle, \langle \iota_2, \gamma_2\rangle, \langle \iota_3, \gamma_3\rangle)]\!]^{\mathcal{S}}$; therefore, $T$ has the match property.

Considering the main tree $T = (\langle \iota, \gamma\rangle, \mathrm{SAND})(T_1, T_2, T_3)$, and considering all the following sub-branches, the $(\langle \iota_1, \gamma_1\rangle, \mathrm{OR})(T_{11}, T_{12}) \neq \varnothing$, $(\langle \iota_2, \gamma_2\rangle, \mathrm{SAND})(T_{21}, T_{22}, T_{23}) \neq \varnothing$ and $(\langle \iota_3, \gamma_3\rangle, \mathrm{SAND})(T_{31}, T_{32}, T_{33}, T_{34}) \neq \varnothing$ as well as the main tree $(\langle \iota, \gamma\rangle, \mathrm{SAND})(T_1, T_2, T_3) \neq \varnothing$, the tree $T$ is considered to have the Global Admissible property.

The design of attack tree $T$ is found to be consistent and correct with respect to the system $\mathcal{Z}$ in its finite transition system domain. Each of the sub-trees have match property except the sub-tree $T_1$, which has the overmatch property. This however does not affect the overall security features of QF since all the sub-branches of $T$ relate to $OP = \mathrm{SAND}$ thus complementing each other. Thus, the attack tree $T$ modelling can be used to further analyze the security features of QF.

## 6. Conclusions

This paper discussed MySANI, the enhanced method in regulating the software used for NAWI within the LM framework in Malaysia. This method is realized by introducing QF, which is used as a secondary information source of LR information of the TOE and secured via public cryptography. The security design is validated by using attack tree modelling where the attack tree is described using formal notation and the correctness validation is achieved via comparison against the finite state transition system. As part of security analysis in LM, this paper also demonstrated the possibility of the formalization design of attack tree before it could be integrated with the AtPT framework analysis. This enables the attack tree to be verified prior to further security risk analysis evaluation. The issue of scalability is not studied in this paper due to the nature of the LR part of NAWI, which is usually very small. However, in the future, if the security objects in this paper are to be implemented for other types of measuring instruments, the scalability issues of the formal method approach might be considered.

**Author Contributions:** Conceptualization, M.A.I. and F.Q.; Funding acquisition, Z.S.; Methodology, N.Z.; Supervision, N.M.; Visualization, Z.S.; Writing—original draft, M.A.I. and N.Z.; Writing—review and editing, F.Q., N.M. and M.U.S. All authors have read and agreed to the published version of the manuscript.

**Funding:** This paper is supported by the Universiti Kebangsaan Malaysia Grant Number: KKP/2020/UKM-UKM/4/3.

**Data Availability Statement:** Not applicable.

**Acknowledgments:** The authors would also like to thank the respected editor and reviewer for their support.

**Conflicts of Interest:** The authors declare no conflict of interest.

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
