# Peer review of "Formalizing Attack Tree on Security Object for MySANi in Legal Metrology"

_systems, doi:10.3390/systems11010049_

Round 1
Reviewer 1 Report
Strengths:
(+) The paper is well-written.
(+) The problem is well-defined.
(+) The literature review is good.
(+) The proposed method is well-explained.
(+) The experiments are convincing.
Weaknesses:
(-) There are English issues.
(-) The introduction must be improved.
(-) The method is not novel enough.
(-) Experimental evaluation must be improved.
==== ENGLISH ====
Some sentences are too long. Generally, it is better to write short sentences with one idea per sentence.
==== INTRODUCTION ====
Contributions should be highlighted more. It should be made clear what is novel and how it addresses the limitations of prior work.
Introduction is too long. Authors should consider moving some content to other sections. The role of the introduction is to motivate the problem, briefly highlight limitations or prior work and give a short overview of the contributions.
==== NOTATIONS ====
Please add a table that summarize all the notations used in the paper.
==== EXPERIMENTS ====
Some additional experiments are required:
- Scalability
Author Response
Attached is the reviewer response file.

Reviewer 2 Report
In the paper, interesting research outcomes are presented. The Authors have proposed Malaysia NAWI Software Inspection for Computer (MyNaSIC), which is an enhanced method of regulating the software used for NAWI within the legal metrology framework in Malaysia. Moreover, the Authors (based on the best practices in IT in metrology) have made the security evaluation of the proposed approach, where the attack model on relevant assets of the security objects has been simulated for the Attack Probability Tree (AtPT). Attack tree design in the AtPT framework has been enhanced using formal notation and the correctness validation is achieved via comparison against the finite state transition system.
The paper is technically sound and describes in a comprehensive way the aim and obtained results. The paper is well organized and written on a good level. However, in my opinion, the Authors’ contribution should be more clearly presented (e.g., in the Introduction section).
Moreover, the paper needs to be improved, and some minor mistakes should be corrected:
a) sometimes there are unnecessary capital letters like in line 16 (“Software”), or in line 25 (“simulated for The”), and sometimes there is a lack of capital letters like in line 30 (“integrity”);
b) in the keywords section there is “and protection” (line 31) that sounds weird (what does the word “protection” refer to?);
c) in line 57 the abbreviation “NAWI” is introduced but it stands for “Non-Automatic Weighing Instruments” not “Non-Automatic Measuring Instruments”,
d) in line 62 the abbreviation “HC” is introduced – what does it stand for?;
e) in line 80 the sentence “To effectively protect an asset, one must understand the harm exposed by an asset.” is not clear to me;
f) in Fig. 2 there is an abbreviation “OIML” – what does it stand for?;
g) in Fig. 3 there is an explanation of the meaning of the red dashed line, but in the figure, there is used another style of dashed line (it looks like a solid line) – in Fig. 4 it is correct;
h) what does the red dashed line in Fig. 5 mean?
i) in line 162 there is “propery” and should be “property”;
j) in line 335 there should be a coma after “i.e.”;
k) in line 417 there is “enrypts” and should be “encrypts”.
Author Response
Attached is the reviewer response file.
